# Ship Berthing Trajectory Cluster based on Variational Inference Improved Mean Shift

Han Xue
College of Navigation
Jimei University
Xiamen, China
imlmd@163.com

Kun Qian
College of Navigation
Jimei University
Xiamen, China
qk1027@163.com

*Abstract*—In order to improve the performance of berthing trajectory clustering in presence of wind resistances, mean-shift is improved for clustering. Its convergence is then proven. Finally, the feasibility of this method is proved by experiments compared with other cluster algorithms. It is shown that this method can improve the clustering effect to a certain extent. The new algorithm is applied to the clustering of ship berthing trajectories to mine the berthing trajectories under different conditions, such as different navigable waters, heading, speed, rudder angle, channel depth, ship size, ship type, safe distance from the dock sideline, other ships in adjacent berths, traffic flow density and special weather.

*Keywords—Trajectory cluster, Mean-shift, Ship birthing, Variational inference*

## I. INTRODUCTION

### A. Research background

The wharf water area is a restricted water area, with heavy traffic, complex situation and frequent collision between ships. Many factors have a significant impact on berthing. With the larger and larger ships, more and more super large container ships are berthing at the wharf. If the prediction is insufficient and the ship is manipulated improperly, it will easily lead to the danger of grounding and collision. Therefore, under different conditions, it is very important to choose the appropriate berthing method and mode to ensure the smooth implementation of berthing operations.

Ship berthing has important theoretical research and engineering practical value because of its complexity. At present, the berthing of ships is mostly completed manually. The maneuverability of large ships is limited, the rudder effect is weakened at low speed, easy to be affected by wind and current. With data mining technology, cluster analysis is carried out on the ship berthing track, and typical berthing mode is constructed. This can help discover ship motion characteristics and behavior patterns, and provide new berthing modes. The data of automatic identification system (AIS) contains a large number of marine traffic characteristics, which can mine the effective and potential information about ship motion law in AIS data. Mining relevant behavior patterns from ship AIS data to assist water safety supervision is of great significance to the increasingly complex maritime traffic safety situation.

### B. Related work

In the area of trajectory clustering, the shortcomings of traditional clustering algorithm in parameter setting and noise recognition was solved[1]. It solved the exclusivity problem of traditional algorithms. By evaluating the historical ship behavior in a given geographical area, Murray applied machine learning technology to infer the commonness of relevant trajectory segments[2]. These commonalities represented the historical behavior patterns corresponding to the possible future behavior of the selected ship. The selected ships were classified into behavior patterns, the trajectories related to the patterns are predicted, and the Gaussian mixture model was used for clustering. Zhang used the noise based on fuzzy adaptive density to cluster the turning points obtained in the preprocessing stage by spatial clustering technology to obtain the turning region[3]. Gao proposed a pattern recognition method of ship maneuvering behavior[4]. T-distributed random neighborhood embedding algorithm was used to reduce the dimension of seven tuple data.

Among the area of ship berthing behavior, Chen developed a berth information extraction system based on 3D Lidar[5]. The principal component analysis method was used to calculate the ship heading and normal vector, and determined the characteristic points of the bow and stern. The segment passing through the point was obtained by region growth. Through visibility analysis, according to the position of the ship relative to the berth, the bow and stern were identified by the similarity between the normal vector of the leg and the ship course. Lee predicted the risk of unsafe berthing speed[6]. Liao used the analysis of environmental constraints, berth point constraints and USV dynamic constraints[7].

Mean shift algorithm was first proposed by Fukunaga and others in 1975[8]. A point iteration updating equation was proposed, which combined the updating equation of standard mean shift and fuzzy mean shift. Robust Mean-Shift can be set based on kernel and nearest neighbor[9]. Since the update rule of root mean square is closer to ,Blurring Mean-Shift the convergence of point iteration is inferred based on Blurring Mean-Shift convergence theorem. Clustering based on density estimation was used for trajectory correlation[10]. The point, center and distance were redefined to extend the mean shift clustering method, and the bandwidth helps to handle different scenarios was adjusted. Mean shift clustering was applied to data base technology to achieve efficient anti pattern discovery in the shortest time[11]. A faster mean shift algorithm was proposed, based on embedded clustering[12]. An attack

detection method based on weighted Euclidean distance mean shift clustering algorithm was proposed[13].

In the study area of variational inference, a variational inference framework was developed for kernel fried tensor, which could associate prediction and prediction with calibration uncertainty estimation on multiple data sets[14]. A variational inference framework called energetic variational inference was introduced[15]. A particle based energetic variational inference scheme first performed particle based density approximation, and then used approximate density in the variational process, which was referred to as "approximation before variation". A variable Bayesian method was proposed to represent the Lévy adaptive regression kernel model of functions with over complete systems[16]. The variational Gaussian approximation was naturally embedded into the EM framework, and the analytical expression of the EM variational reasoning algorithm was derived[17]. A finite inverse Dirichlet hybrid model using variational inference for unsupervised learning was proposed[18]. An incremental algorithm with component segmentation method for local model selection mafe the clustering algorithm more efficient. Variational inference was used to solve Bayesian problems, and transformed Bayesian reasoning into optimization tasks[19]. A general variational inference algorithm automatic differential variational inference was introduced into Bayesian sliding inversion, and compared with the classical metropolis Hastings sampling method. The problem of joint estimation was solved by parallel variable Bayesian inference algorithm, and an adaptive mesh pruning mechanism was designed[20]. The Laplace density function was expressed as the superposition of infinite Gaussian distributions by introducing new latent variables, and the variational expectation maximization (EM) algorithm was used to learn the parameters[21].

Like Maximum Likelihood Estimation, EM is easy to fall into the local extreme value of the model, and EM needs to manually determine the number of Gaussian components K and select the initial value of the parameters of each Gaussian component Improper parameter selection is prone to data over fitting or slow model convergence At the same time, both EM and MLE cannot use the existing prior information, and are sensitive to the initial value of each Gaussian component in the model.

## C. Contributions

(1) A new clustering algorithm is designed.

(2) Different berthing methods in presence of wind resistances are discussed, such as offshore wind in the bow direction, offshore wind in the stern direction, onshore wind in the bow direction and onshore wind in the stern direction.

(3) The new algorithm is applied to the clustering of ship berthing trajectories to mine the berthing trajectories under different conditions, such as different navigable waters, heading, speed, rudder angle, channel depth, ship size, ship type, safe distance from the dock sideline, other ships in adjacent berths, traffic flow density, special weather, etc. Different berthing methods include slowing down and berthing, parallel berthing, turning round and berthing manipulation, etc.

## II. PRELIMINARIES

### A. Ship berthing types

Under different circumstances, ships adopt different berthing maneuvering methods. For example, the is direct approaching berthing method shown in Fig. 1, and the turning round berthing method is shown in Fig. 2.

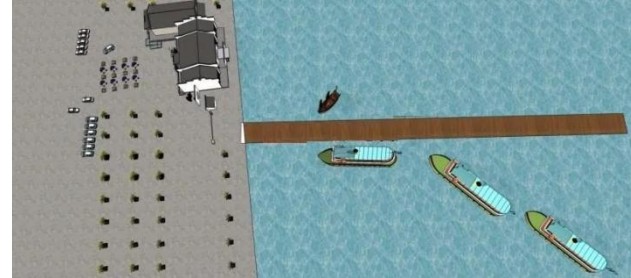

Fig. 1. Direct approaching berthing method.

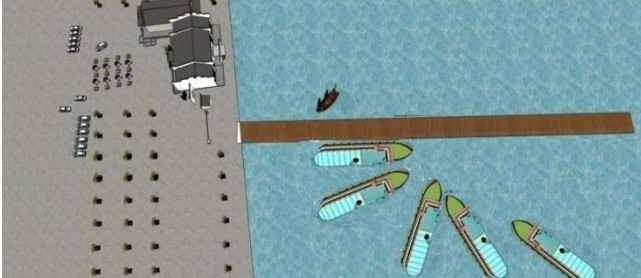

Fig. 2. Turning round berthing method.

Onshore wind is a kind of wind that forces a ship to squeeze against the shoreline of a wharf. Blowing the wind makes the ship approach the wharf with the wind, and the speed is difficult to control. When leaving the wharf against the wind, the direction of the ship is difficult to grasp. If there is a slight negligence, it is very easy to cause an average accident.

Offshore wind is a kind of wind that blows away, and forces the ship to leave the wharf. The condition of wind blowing from a wharf or berth to a channel. There are front splay blowing wind, positive transverse blowing wind and rear splay blowing wind. When the empty ship berths in the static water port with the blowing wind of level 5 ~ 6, the closing angle should be large because the closing component without flow can offset the blowing force of the wind.

With offshore wind in the bow direction, when the bow is 2-3 times from the front end of the berth, the captain is the key time to control the ship's position. Adjust the ship's position and heading so that the bow points to the front end of the berth. Select an appropriate berthing angle according to the wind direction, and minimize the intersection angle between the ship body and the wind direction, the smaller the better.

With offshore wind in the stern direction, if there is enough circulating water in the outer barrier of the berth, the course can be gradually adjusted when it is 2 times from the bow position of the berth, so that the bow of the ship vertically points to the middle position of the berth, so as to reduce the wind side angle and weaken the impact of the wind on the ship.

With onshore wind in the bow direction, due to the wind blowing, the cross distance before berthing should be appropriately increased to not less than 3 times the ship width. The greater the wind, the greater the corresponding span.

With onshore wind in the stern direction，select the appropriate berthing angle according to the wind direction. Try to make the angle between the hull and the wind direction as small as possible. As the intersection angle between the wind direction and the wharf trend gradually increases, the bow direction gradually moves backward from the front end of the berth. When the wind direction is perpendicular to the wharf trend, the bow points to the position slightly forward in the middle of the berth.

## 3. MAIN RESULTS

### A. Architecture of the new algorithm

This paper uses the variational inference based expectation maximum algorithm (VIEM)，as given in Algorithm 1.

---

**Algorithm 1** VIEM-Mean-shift.

**Input:** Problem description and algorithm parameters.

**Output:** The optimal clusters.

1: **for** $m$=1, 2, …, $M$ **do**

2: **for** $s$=1, 2, …, $C$ **do**

3: Computer the distance between each sample and each cluster center.

4: **end for**

5: **end for**

6: **for** t=1,2,…,T **do**

7: Update $q(x,\theta_t)$

8: Calculate ELOB$(q,x,\theta_t)$

9: if |ELOB$(q,x,\theta_t)$-ELOB$(q,x,\theta_t)$|<$\varepsilon$, goto step 11.

10: **end for**

11: Take the largest $q_{ms}$ and assign the $m$ sample to the s-th cluster.

12: Select a sample in each cluster and draw a circle with this sample as the center and bandwidth as the radius.

13: Calculate the average mean shift value of the vector from this sample to other samples in the same cluster.

14: The center of the circle is moved along the mean shift value of the vector, and is the new center of the circle.

15: Repeat steps 14-15 until the termination condition is satisfied.

---

The process of the algorithm is shown in Fig. 3.

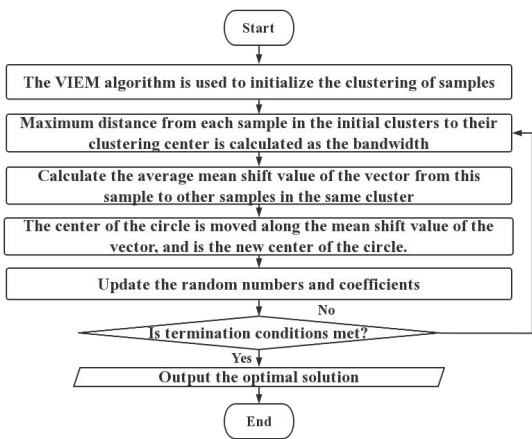

Fig. 3. Architecture of algorithm.

## III. EXPERIMENTS AND ANALYSIS

### A. Instance introduction

Three datasets are used. One is Xiagu Ferry in Dongdu Port. Dongdu port 0-4# berth is planned to be a cruise berth area. The length of berth 0# of the international cruise center wharf is 419m and the berthing capacity is 150000. The length of Berth 1# of the international cruise center wharf is 324m and the berthing capacity is 80000. The length of berth 2# of the international cruise center wharf is 346m and the berthing capacity is 80000. The length of Berth 3# of the international cruise center wharf is 241.3m, Berthing capacity 3000.

Under the influence of typhoons in summer and autumn, the maximum wind speed reaches level 12, and the cold and strong wind in winter can reach level 7-8. The nautical chart in Xiagu Ferry is shown in Fig. 4.

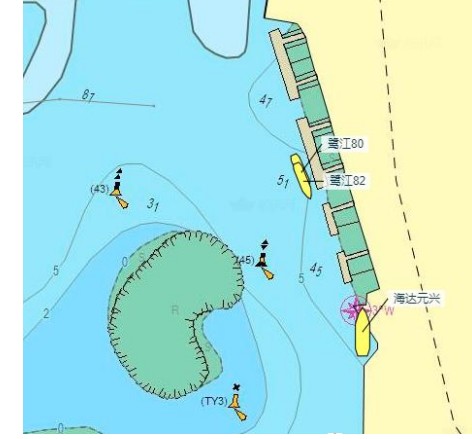

Fig. 4. Nautical chart in Xiagu Ferry.

The wharf of Houshi Power Plant located at 24 °18'30"n, 118 °07'50" E. It is 1.2km away from the coast. The whole wharf is arranged as a jetty, starting from the Cape of xiaoaotou, and its axis extends 38° north by east to the sea. The wharf has a total length of 590m and a width of 24.0m, and can dock one 100000 ton bulk carrier or two 75000 ton bulk carriers. The ship turning area in the port area is located at the wharf apron. The turning area is 650m x 600m, and the design bottom elevation is - 14m.

In Houshi port area, southwest wind prevails in summer, and will be attacked by typhoon from July to September every year. The wind in the port area before and after the typhoon can reach level 6-7. Northeast wind prevails in winter and has a great impact. The nautical chart in Houshi Power plant wharf is shown in Fig. 5.

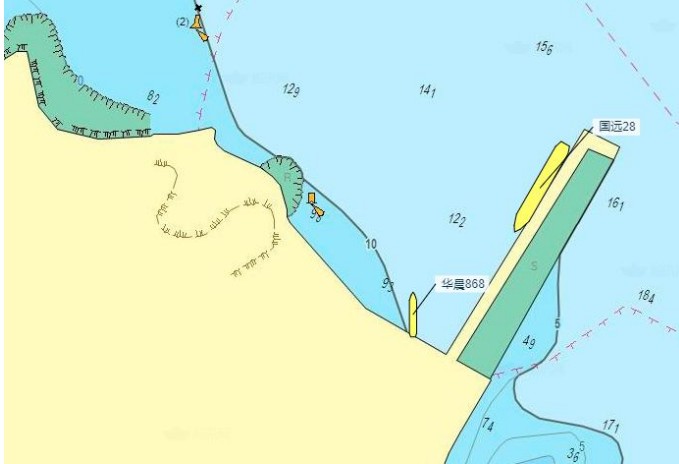

Fig. 5. Nautical chart in Houshi Power plant wharf.

Houshi No. 3 Wharf locates at 24°21'30 "N, 118°05'54" E, in the southwest of Xiamen Island, north of Taiwu mountain, west of Lantau waterway, across the sea from Lantau, Wuyu and Jinmen Islands. The total length of the wharf coastline is 422m. The wharf trend is 126 ° / 306 °. The turning water area is arranged in an oval shape. The long axis is 867m .The short axis is 578m, and the design bottom elevation is - 16m. The construction scale of Houshi No. 3 wharf is 150000 t, and the structure is designed as 200000 t with a total area of about 360000 m².

In Houshi Port, the annual wind is east-north-east with a frequency of 22.97%, followed by northeast with a frequency of 11.7%. The seasonal variation of wind direction is obvious: in autumn, winter and spring, it is dominated by east-north-east with the frequency of 29.5%, 33.8% and 21.3% respectively. In summer, it is dominated by south-south-west with the frequency of 16.9%. Generally speaking, the wind direction of this port area is mainly NE and east-north-east, with an annual occurrence frequency of 35.48%, while the wind direction of SW and SSW mainly occurs in summer, and the cumulative annual occurrence frequency is only 12.46%. According to the number of days with wind speed ≥ 10.8m/s (Level 6 wind), the cumulative number of days with strong wind in the year is 19 days, and the cumulative number of days with wind force greater than level 6 is 2.8 days. The axis orientation of the approach channel in Houshi Port is 327 ° ~ 147 °. The intersection angle between NE and SW winds and the channel axis is 78 °. The wind pressure difference angle is large and close to the cross wind. The influence of wind direction and wind speed on ship handling must be fully considered when entering and leaving the waters near the wharf. The  nautical chart in Houshi No. 3 wharf is shown in Fig .6.

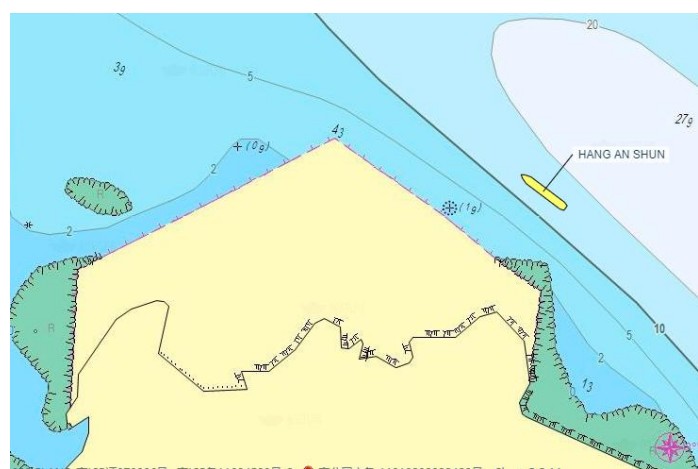

Fig. 6. Nautical chart in Houshi No. 3 wharf.

The wind information on Dec. 31 is listed in Table 1.

TABLE I.    WIND INFORMATION ON DEC. 31.

| Time | Wind speed (m/s) | Maximal wind speed (m/s) | Wind-force |
|------|------------------|--------------------------|------------|
| 06.20 PM | 13.70 | 17.10 | 6.00 |
| 06.30 PM | 14.00 | 16.80 | 7.00 |
| 06.40 PM | 13.40 | 16.10 | 6.00 |
| 06.50 PM | 13.90 | 17.40 | 7.00 |
| 07.00 PM | 14.70 | 18.70 | 7.00 |
| 07.10 PM | 13.90 | 17.10 | 7.00 |
| 07.20 PM | 13.90 | 16.70 | 7.00 |
| 07.30 PM | 13.50 | 16.10 | 6.00 |
| 07.40 PM | 14.10 | 17.20 | 7.00 |
| 07.50 PM | 14.10 | 17.60 | 7.00 |
| 08.00 PM | 13.70 | 16.60 | 6.00 |
| 08.10 PM | 14.10 | 18.50 | 7.00 |
| 08.20 PM | 13.60 | 17.00 | 6.00 |
| 08.30 PM | 13.80 | 16.70 | 6.00 |
| 08.40 PM | 13.60 | 16.30 | 6.00 |
| 08.50 PM | 13.60 | 17.20 | 6.00 |
| 09.00 PM | 13.50 | 16.70 | 6.00 |
| 09.10 PM | 13.80 | 16.60 | 6.00 |
| 09.20 PM | 13.70 | 17.20 | 6.00 |
| 09.30 PM | 13.00 | 15.70 | 6.00 |

The wind field is shown in Fig .7.

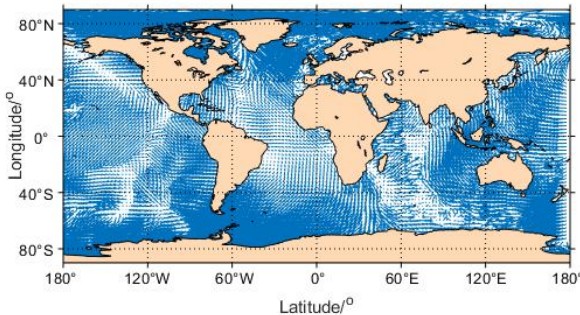

Fig. 7. Wind field.

The ship berthing trajectory in Xiamen, including (MMSI), time, longitude, latitude, course over ground (COG), speed over ground (SOG) is listed in Table 2.

TABLE II.       SHIP BERTHING TRAJECTORY ON AUG. 14, 2021.

| MMSI | Time | Longitude/° | Latitude/° | COG/° | SOG/knot |
|------|------|-------------|------------|-------|----------|
| 412751910 | 19:16:22 | 117.981252 | 24.406588 | 330.0 | 0.2 |
| 412751910 | 19:17:22 | 117.98124 | 24.406607 | 125.9 | 0.1 |
| 412751910 | 19:17:37 | 117.981253 | 24.406593 | 139.8 | 0.2 |
| 412751910 | 19:18:35 | 117.981303 | 24.406537 | 175.9 | 0.1 |
| 412751910 | 19:19:36 | 117.981265 | 24.40655 | 333.4 | 0.2 |
| 412751910 | 19:20:21 | 117.981242 | 24.406582 | 353.9 | 0.2 |
| 412751910 | 19:20:50 | 117.981248 | 24.406595 | 34.4 | 0.1 |
| 412751910 | 19:21:05 | 117.981257 | 24.406597 | 65.9 | 0.1 |
| 412751910 | 19:22:06 | 117.981257 | 24.406587 | 197.2 | 0.1 |
| 412751910 | 19:22:21 | 117.98126 | 24.406585 | 134.7 | 0.0 |
| 412751910 | 19:22:36 | 117.981258 | 24.406588 | 346.3 | 0.0 |
| 412751910 | 19:23:35 | 117.981255 | 24.406607 | 107.3 | 0.0 |
| 412751910 | 19:24:51 | 117.981262 | 24.406605 | 299.3 | 0.1 |
| 412751910 | 19:25:19 | 117.98124 | 24.406618 | 305.0 | 0.2 |
| 412751910 | 19:25:21 | 117.98124 | 24.406618 | 304.8 | 0.2 |
| 412751910 | 19:26:52 | 117.981243 | 24.406635 | 87.2 | 0.2 |

## B. Results of different datasets

The ship berthing trajectory clustering result in Xiagu Ferry is shown in Fig. 8.

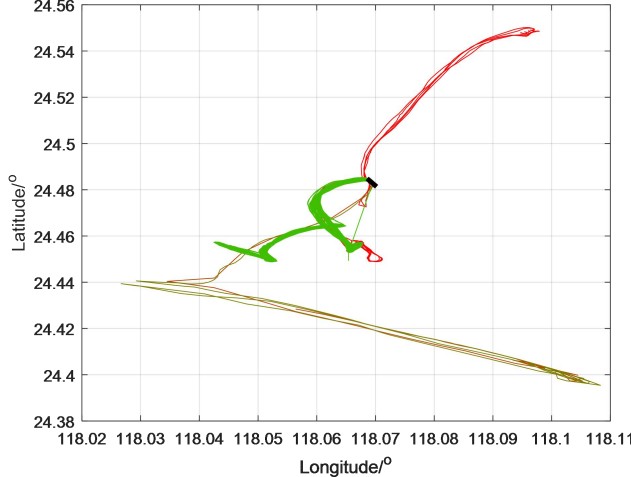

Fig. 8. Berthing trajectory clustering in Xiagu Ferry.

The berthing trajectory clustering in Xiagu Ferry plotted in nautical chart is shown in Fig. 9.

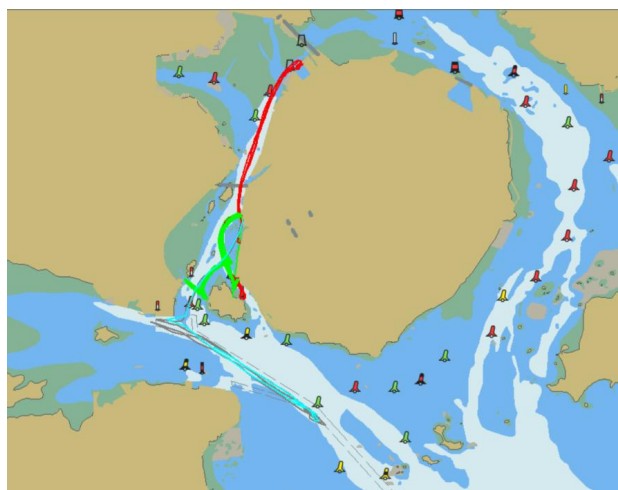

Fig. 9. Berthing trajectory clustering in Xiagu Ferry plotted in nautical chart.

The ship berthing trajectory clustering result in Houshi Power plant wharf is shown in Fig. 10.

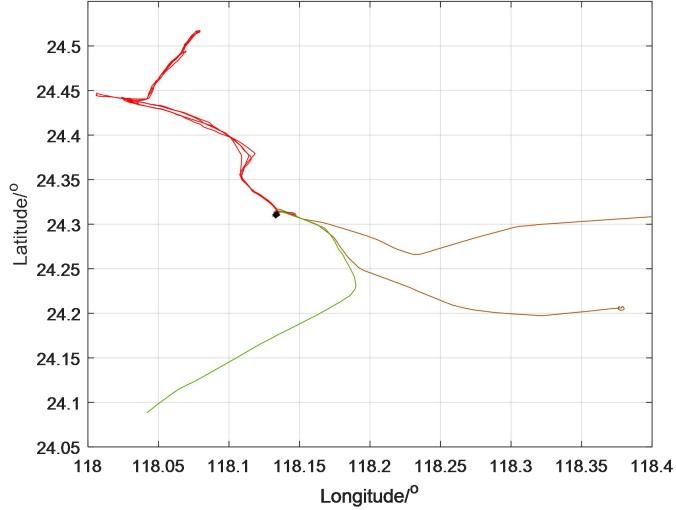

Fig. 10. Berthing trajectory clustering in Houshi Power plant wharf.

The ship berthing trajectory clustering result in Houshi Power plant wharf plotted in nautical chart is shown in Fig. 11.

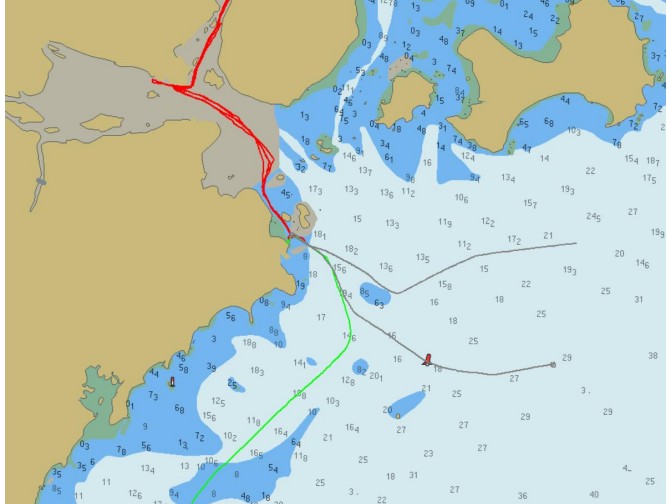

Fig. 11. Berthing trajectory clustering in Houshi Power plant wharf.

The ship berthing trajectory clustering result in Houshi No. 3 wharf is shown in Fig. 12.

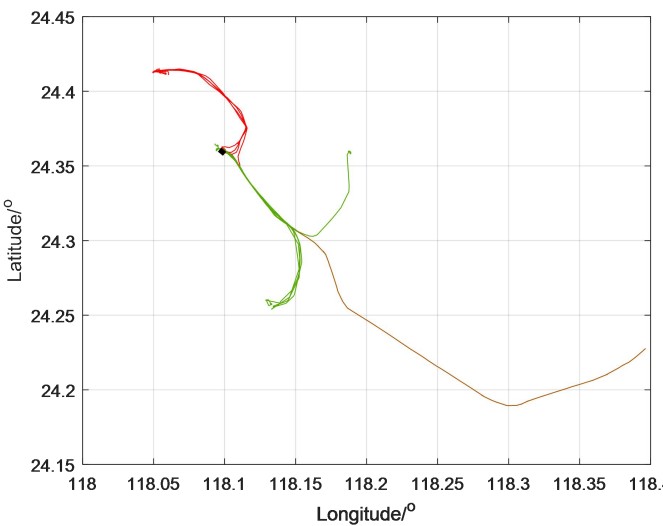

Fig. 12. Berthing trajectory clustering in Houshi No. 3 wharf.

The ship berthing trajectory clustering result in Houshi No. 3 wharf plotted in nautical chart is shown in Fig. 13.

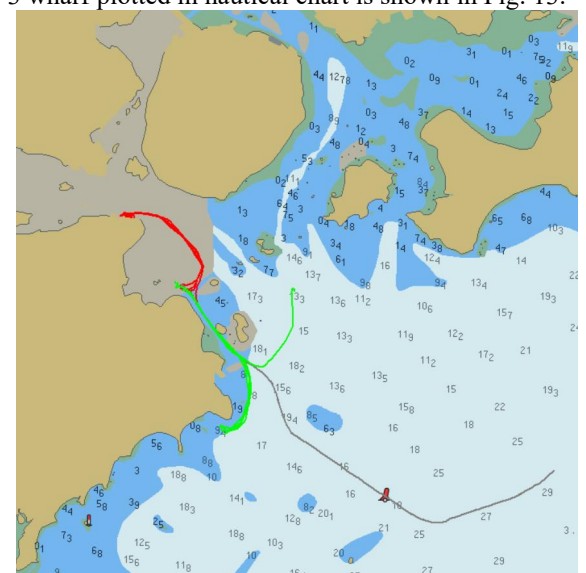

Fig. 13. Berthing trajectory clustering in Houshi No. 3 wharf.

The ship berthing trajectory clustering result in Xiagu Ferry is shown in Fig. 14.

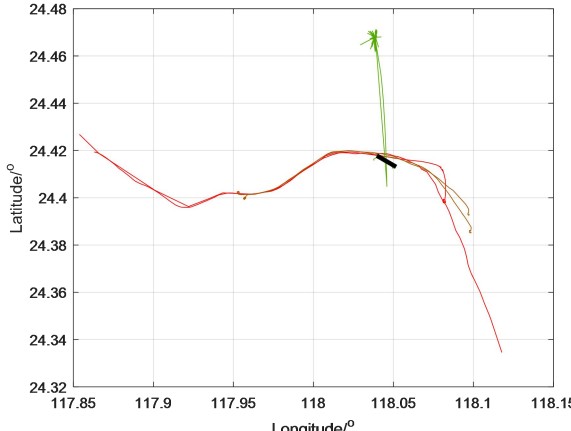

Fig. 14. Berthing trajectory clustering in Xiagu Ferry.

The ship berthing trajectory clustering result near modern wharf is shown in Fig. 15.

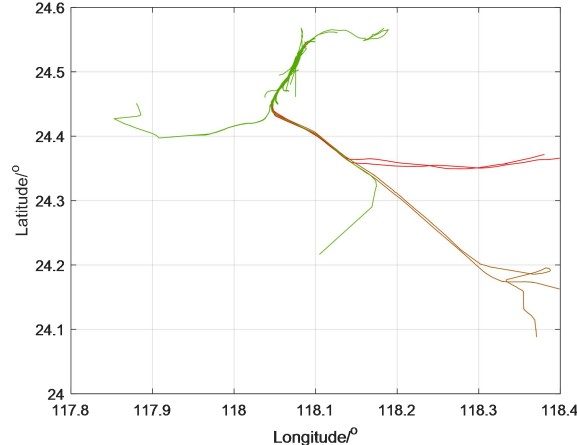

Fig. 15. Berthing trajectory clustering in modern wharf.

The experimental results show that the ship trajectory clustering model constructed in this paper are suitable for ship trajectory mining. It can classify the ship berthing traffic flow in the wharf area, and can obtain a better trajectory clustering effect.

## C. Comparison of different algorithms

The proposed algorithm is compared with differential evolution clustering algorithm, such as differential evolution clustering algorithm[22] and mean shift algorithm. The clustering results near Guanhaiyuan wharf are shown in Fig. 16.

| Algorithms | Best | Worst | Average | Best | Worst | Average | Best | Worst | Average |
|---|---|---|---|---|---|---|---|---|---|
| Mean-shift | 1.1875 | 1.2360 | 1.1904 | 1.3450 | 1.3907 | 1.3670 | 0.9045 | 1.0654 | 0.9296 |
| VIEM-Mean-shift | 0.9546 | 0.9962 | 0.9745 | 1.1166 | 1.1570 | 1.1296 | 0.7641 | 0.8034 | 0.7907 |

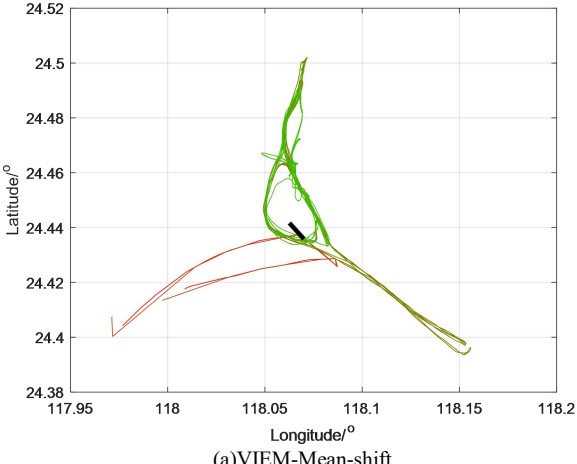

(a)VIEM-Mean-shift

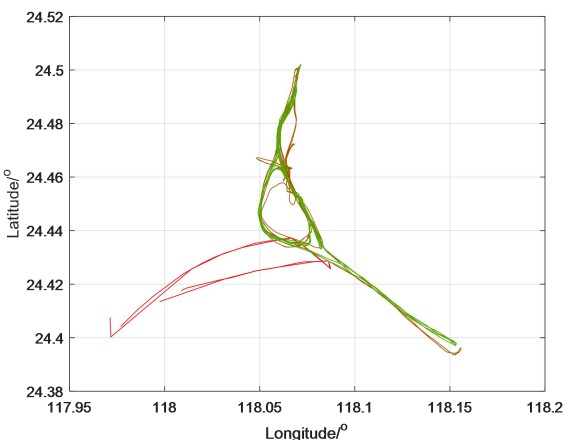

(b) differential evolution clustering algorithm

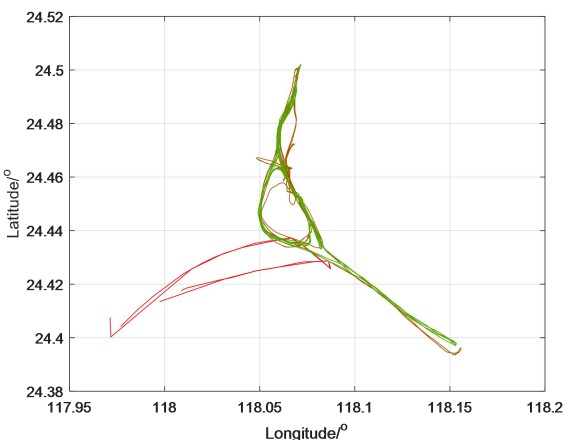

(c)Mean-shift

Fig. 16. Clustering results near Guanhaiyuan wharf.

Fig. 16 shows that compared with K-means algorithm and differential evolution clustering algorithm, the new algorithm has higher calculation accuracy.

Table 3 lists the comparison of best costs.

TABLE III. BEST COST COMPARISON OF DIFFERENT ALGORITHMS.

| Dataset | Xiagu Ferry | Houshi Power plant wharf | Houshi No. 3 wharf |
|---|---|---|---|

### D. Discussion

Houshi Post has southwest wind in summer and northeast wind in winter. The wharf of Houshi Power Plant has little room for maneuver and is greatly affected by wind and waves. When the southwest wind superimposes the rising water, if the ship is close to the upper line of buoy 105, it is very easy to be pressed by the wind on buoy 105 or the shoal outside the channel.. Since the heavy-duty ship operates slowly, the effects of wind-induced drift should be fully considered. Houshi wharf and channel are located in open waters, basically without shelter, and there are often strong winds and waves.

According to the rose chart of wind, when the ship enters the port, strong wind and strong wave happen to be on the starboard side. In order to reduce the risk of berthing, it is usually to dock on the port side.Turn around when the ship is crossing the wharf angle, adjust the rotation speed of the large ship and the transverse distance from the wharf according to the wind and current pressure in the harbor basin.

As long as the water depth of the harbor basin is sufficient, it can enter the berth during the ebb tide period. During spring tide, the flow velocity is large, and a slight angle in berthing can form a large flow pressure, which is difficult to control the ship. It is necessary to choose a period with slow flow velocity to ensure berthing safety. Under general wind flow conditions, parallel berthing mode is adopted. During berthing, the ship moves laterally. When it is 1 times the ship width from the berth, a ship enters the berth slowly at a small berthing angle. When it is half the ship width from the berth, it enters the berth in parallel.

## IV. CONCLUSION AND FUTURE WORK

Based on the improved clustering algorithm with new bandwidth selection mechanism and convergence analysis, the clustering method of ship berthing trajectory is studied, and verified by taking the AIS data of Xiamen port in August 2021 as an example. The results show that the algorithm can greatly improve the efficiency of trajectory clustering, and provide technical support for further research on ship motion characteristics, behavior patterns and real-time anomaly detection of ship trajectory.

In future, the influence of surrounding dynamic ships will be also taken into account for better ship berthing trajectory clustering.

ACKNOWLEDGMENT

This work was supported in part by the National Natural Science Foundation of China (No. 52201411) and Natural Science Foundation of Fujian Province (No. 2023I0019).

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
