# OpenReview forum: "Ship Berthing Trajectory Cluster based on Variational Inference Improved Mean Shift"
_IEEE.org/ICIST/2024/Conference — IEEE ICIST 2024 Conference Submission_

### Official Review · Reviewer_9Bq8 · 2024-08-25
**minor repair**

**Rating:** 8
**Confidence:** 3

**Review:**

1. In the contribution section, the author should add comparisons with existing literature。
2. I would suggest the authors polish the language of this paper carefully.

---

### Official Review · Reviewer_f19p · 2024-08-29
**Review on Ship Berthing Trajectory Cluster based on Variational Inference Improved Mean Shift**

**Rating:** 6
**Confidence:** 3

**Review:**

1. The readability of this paper needs improvement. The explanation of the corresponding abbreviation should be given after the first appearance of the abbreviation. Please give the explanations of EM LME, and ELOB where they appear for the first time. Besides, a period is missing in the last paragraph of Section I B.
2. The contribution (1) A new clustering algorithm is designed. It is too brief and doesn't explain the contribution clearly.
3. In the Algorithm 1, step 15 is: Repeat steps 14-15 until the termination condition is satisfied. I guess there is a mistake.
4. In the paragraph below Fig. 4, ''The wharf of Houshi Power Plant located at 24 °18'30"n,'' n should be an uppercase.
5. Comparison of different algorithms is not convincing. According to Fig.16, the comparison is not clear enough. More explanation or a table is needed.
6. Table III is very confusing. One number is split in 2 or 3 lines and it seem to be 2 or 3 numbers.
7. What are the differences between Fig. 8 and Fig. 14?

---

### Official Review · Reviewer_aLmK · 2024-08-30
**comment**

**Rating:** 7
**Confidence:** 4

**Review:**

1. The description of the figures in the article is not detailed enough, legends should be added. For example, Figure 8.
2. It is recommended to add a paragraph of explanation in Chapter 3, Part B, otherwise the reader will not understand until the last paragraph of Part B.
3. Figure 16 can hardly indicate that the method proposed in this paper has higher accuracy.

---

### Decision · Program_Chairs · 2024-09-06

Accept (Oral)